# Fluoropolymer: A Review on Its Emulsion Preparation and Wettability to Solid-Liquid Interface

**DOI:** 10.3390/molecules28020905

**Published:** 2023-01-16

**Authors:** Lei Liang, Tao Wen, Jun Xin, Chao Su, Ke Song, Wei Zhao, Hongwu Liu, Gui Su

**Affiliations:** Geological Exploration and Development Research Institute, CNPC Chuanqing Drilling Engineering Company Limited (CCDC), Chengdu 610051, China

**Keywords:** fluoropolymer, emulsion polymerization, wettability, solid–liquid interface, nanomaterials, enhancing oil recovery

## Abstract

In the preparation of a superamphiphobic surface, the most basic method is to reduce the surface free energy of the interface. The C—F bond has a very low surface free energy, which can significantly change the wettability of the solid–liquid interface and make it a hydrophobic or oleophobic, or even superamphiphobic surface. Based on the analysis of a large number of research articles, the preparation and application progress in fluoropolymer emulsion were summarized. After that, some corresponding thoughts were put forward combined with our professional characteristics. According to recent research, the status of the fluoropolymer emulsion preparation system was analyzed. In addition, all related aspects of fluoropolymer emulsion were systematically classified in varying degrees. Furthermore, the interaction between fluoropolymer structure and properties, especially the interaction with nanomaterials, was also explored. The aim of this review is to try to attract more scholars’ attention to fluorocarbon interfacial materials. It is expected that it will make a certain theoretical and practical significance in the preparation and application of fluoropolymer.

## 1. Introduction

In order to adapt to the natural environment, organisms have evolved waterproof ability in some tissues and organs, such as the flowers and leaves of lotus [1], the feet of water striders [2], and the feathers of kingfishers [3]. With this inspiration, people began to explore the production mechanism of the superhydrophobic phenomenon. In the preparation of a superhydrophobic surface, reducing its surface free energy is the core problem when changing its wettability [4]. Changing the roughness of the interface and coating the interface with low surface free energy materials are the most common methods [5]. The C—F bond has a very low surface free energy. The wettability of the solid–liquid interface can be significantly changed by using fluorinated compounds, which can achieve a hydrophobic or super-amphiphobic effect [6,7]. However, the dispersion of fluorocarbons in the preparation process has become a difficult problem due to the low free energy of fluorocarbons. Especially in polymerization, the type and cost of materials hindered the development in the fields [8,9]. In recent years, the development of the fluorine industry has promoted the process of fluoropolymer. Abundant fluorinated monomers have made corresponding breakthroughs in the research of fluoropolymers. In the preparation of fluoropolymer, fluorinated acrylic monomers are widely used [10,11]. In the meantime, the diversification of emulsifiers and emulsion polymerization has also effectively solved the dispersion problem of fluoropolymer in the system [12].

Fluoropolymer emulsion is a kind of polymer obtained by emulsion polymerization. Under the action of an emulsifier, the polymer particles can be dispersed evenly to form a stable emulsion. According to the dispersing medium, it can be divided into the water-based emulsion and oil-based emulsion [13,14]. Fluoropolymer has good thermal stability [15], good flame retarding [16], anti-oxidation [17], and anti-aging effect [18]. It has good chemical inertness and strong adaptability to the environment. Therefore, it is widely used in aerospace [19], optical electronics [20], textile coatings [21], and other fields. At the same time, fluoropolymers also have significant defects, such as poor solubility in organic solvents, and synthesized fluoropolymers generally have high crystallinity [22,23]. This makes the synthesis and processing more difficult. To overcome these difficulties, emulsion polymerization has been used to increase the dispersity of the synthesized fluoropolymer in the reaction system [24]. In recent years, resource shortage, environmental pollution, and other issues have gradually received the attention of society. To meet the growing and gradually diversified social needs, the research and development of environmentally friendly fluoropolymer [25,26], especially the synthesis and application of fluoropolymer emulsion, is an inevitable trend in the development of organic fluorine materials [27]. The harm of water-based emulsion to the environment is far less than that of oil-based. With the enhancement of people’s environmental awareness, the development and utilization of waterborne fluoropolymer emulsion has attracted more and more attention [28,29,30]. Fluoropolymer emulsion with water basis has great advantages. On the one hand, it can avoid environmental pollution caused by the volatilization of organic solvents. On the other hand, it also can reduce the production and cost of products, and greatly reduces the harm of organic solvents.

After the problems were solved, scholars synthesized many fluoropolymer emulsions with different structures and properties. Fluoropolymer emulsion has been widely used in coating materials [31], the textile industry [32], building materials [33], the petroleum industry [34], medical materials [35,36], and especially in the combination of nanomaterials and applications [37]. It has become the research focus of many scholars. In addition, the development direction of various fields in the international community is towards the sustainable development of energy conservation and environmental protection. In addition to the consideration of some new structures and new methods, the safety of materials becomes more important than the superiority of the performance [38,39]. Researchers have made headway in nano-coating materials by using nanomaterials such as preparing nano-polymer emulsion, grafting modification, or by enhancing the efficiency of materials. Fluoropolymers with different structures and their combination with different nanomaterials can significantly improve the wettability of a solid–liquid interface [40]. In the process of oil and gas development, the wettability of hydrophilic and oleophilic reservoirs can be reversed and transformed into hydrophobic, even amphiphobic ones. Therefore, this can reduce migration resistance, increase the flowback efficiency, and enhance oil and gas recovery.

Through the analysis of the preparation and application of a large number of fluoropolymer emulsions, this article summarized and analyzed the present situation of fluoropolymer emulsion preparation systems, including the polymerization monomers, polymerization methods, emulsification systems, and initiating systems. Combined with our professional characteristics, the research in this field was summarized and the corresponding thinking was put forward. The structure and properties of fluoropolymers were also explored, especially the interaction between fluoropolymers and nanomaterials. The aim is to attract more scholars’ attention to fluorocarbon interfacial materials. It is expected that this will play a certain theoretical and practical part in the preparation and application of fluoropolymer emulsions.

## 2. Fluorinated Monomers

The most commonly used reactions in polymerization are free radical copolymerization and condensation polymerization. The reaction requires at least two functional groups [41]. Therefore, the monomer in the preparation is mostly a fluorinated group on one side, and the other side has a reactive functional group. The chemical structure and composition of segments determines the performance of fluorinated polymers, while the type and number of active functional groups in monomers control the direction of polymerization [42]. Therefore, it is very important to select suitable monomers for emulsion polymerization in many fluorinated monomers [43]. According to the different active functional groups in the monomers, the commonly used fluorinated monomers can be divided into several types (Figure 1).

### 2.1. Alkene (Alkyne) Fluorinated Monomers

Fluorinated monomers containing double or triple bonds are mainly fluorinated alkenes (alkynes) and nitriles (Figure 1a). The fluorinated monomers commonly used are mainly fluoroalkenes containing C2~C4 chains, such as vinyl fluoride (VF), vinylidene fluoride (VDF), trifluoro ethylene chloride (CTFE), tetrafluoroethylene (TFE), hexafluoropropylene (HFP), etc. [44,45]. However, in this kind of structure, the reaction activity of the fluorinated group connected with a double (triple) bond is lower than that of the nonfluorine monomer due to its strong induction effect [46].

### 2.2. Acrylic Acid (Ester) Fluorinated Monomers

Similar to the alkenes (alkynes) fluorinated monomers, fluorinated acrylic acid (ester) monomers are free radical copolymerizations of double bonds (Figure 1b). The difference is that the introduction of carboxyl and ester groups enhances the reactivity and dispersion of the monomers in the system to different degrees, which can improve the adhesion on the substrate, and enhance the curing ability and crosslinking ability of the materials [47]. The most widely studied is the use of fluorinated acrylate monomers. The fluorinated groups in the side chain of fluorinated acrylate polymers play a shielding role in the main chain and internal structure, and still have stable physicochemical properties at a low fluorine content. The fluorinated acrylate polymers after chemical modification can not only inherit the excellent properties of acrylate itself, but also have the advantages of high stability specific surface physical properties and photochemical properties [48,49].

### 2.3. Fluorinated Aromatic (Heterocyclic) Monomers

The electron density of aromatic (hetero) ring compounds is high, which has a greater impact on the surrounding chemical environment (Figure 1c). They have stronger adsorption than ordinary monomers. They can change their solubility in polymers and greatly reduce the crystallization performance of final polymers [50]. These monomers are commonly used as intermediates in fluoropolymers, such as the reaction core of core–shell emulsion polymerization, the branching chain for graft copolymerization, etc. [51].

### 2.4. Others

The above types of fluorinated monomers are polymerized as the center of the addition reaction. Moreover, the elimination of small molecule polycondensation is more accurate in fluoropolymer emulsion (Figure 1d). Unlike addition polymerization, polycondensation mainly relies on the elimination of the end groups in monomers [52]. Generally, a monomer has two or more easily eliminated functional groups, such as hydroxyl, amino, halogen atoms, etc. The advantage of this kind of monomer is that the reaction selectivity is very strong, which is conducive to the design of the specific molecular structure. The disadvantage is that the fluorinated monomer with multiple substituent groups is generally very expensive. In the reaction process, a lot of heat is released, and a lot of energy is consumed to neutralize the energy released from the reaction [53,54].

In addition to the above types of fluorinated monomers, there are also fluorinated monomers such as fluorinated phosphates, fluorinated olefin sulfonates, fluorinated chlorosilanes, etc. [55]. These monomers act as emulsifiers while polymerizing, which can reduce or remove the emulsifiers. The disadvantages are that they are difficult to prepare and retain, and the use cost is high. However, they have great advantages in structural design and reaction stability.

## 3. Fluoropolymer and Emulsion Polymerization

For emulsion polymerization, we generally divide according to the reaction characteristics or the interaction in the reaction process. From the reaction mechanism, polymerization can be divided into free radical polymerization, ion polymerization, and step-by-step polymerization [56]. Additionally, it can be divided into homogeneous polymerization and heterogeneous polymerization according to the compatibility of the reaction system.

The preparation methods can be divided into conventional and unconventional emulsion polymerization. Based on conventional emulsion polymerization, the development and applications of unconventional emulsion polymerization are very extensive. The synthesized emulsion has a stable performance and can meet different application conditions. Next, we compare several different aggregation methods combined with their application examples, and analyze the application scenarios and their respective advantages and disadvantages.

### 3.1. Conventional Emulsion Polymerization

The conventional emulsion polymerization is included with polymerize monomer, water/oil phase, initiator, and emulsifier in the reaction system. This method is easy to operate and with low cost, but the emulsifying performance of the system is not stable [57]. Most of the reactions were polymerized with fluorinated acrylate monomers, and fluoropolymer emulsions with different properties were obtained in different emulsifying systems and initiating systems.

Chen et al. [58] used cationic and non-ionic composite emulsifying systems, and the self-crosslinking copolymer emulsion containing fluorinated acrylate/acrylic eighteen alkyl ester had good crosslinking properties at very low concentrations. Lineman et al. [59] obtained an average size of 191 nm fluoropolymer emulsion using a solid initiator, twelve alkyl three methyl ammonium bromide, as an emulsifier. Under the same structure, nanoemulsion has a better performance. Deng et al. [60] synthesized two block fluoropolymer emulsions with perfluoroacrylate and twelve alcohol acrylate as raw materials. Cheng et al. [61] prepared three fluorocarbon copolymer emulsions with acrylic perfluoroalkyl ester, methyl methacrylate, and butyl acrylate as raw materials. Compared with copolymer emulsion, terpolymer emulsion had a lower concentration. It can achieve a better hydrophobic effect under low temperatures. Zhou et al. [62] used emulsion polymerization to synthesize a fluoropolymer emulsion. After treatment, the contact angle between the fabric and water can reach 142°, which has a good hydrophobic effect.

It is obvious that fluoropolymer is mainly used for its strong hydrophobic ability. Therefore, in conventional emulsion polymerization, besides the variability of initiator and emulsifier, the choice of monomer is also diverse. With graft copolymerization, the characteristics of various monomers can be integrated into one, and fluoropolymers with superior properties can be obtained.

### 3.2. Fine/Mini Emulsion Polymerization

Fine/mini emulsion polymerization shows great advantages over conventional emulsion polymerization. Unlike conventional emulsion polymerization, the nucleation mechanism of micelle in fine/mini emulsion polymerization is monomer droplet nucleation. The nucleated monomer droplet can be regarded as an independent microreactor, in which the monomer is directly polymerized and transformed into nanoparticles [63]. In this way, the premature migration and diffusion of monomers to the aqueous phase can be avoided resulting in the hydrolysis of oligomer and termination of polymerization.

Fine and mini emulsion polymerization is not very different from the reaction mechanism. However, their nucleation size distribution is different. Fine emulsion polymerization is a thermodynamically unstable system than mini emulsion polymerization. Fine emulsion polymerization usually uses C12~C14 alcohols or alkanes as a stabilizer [64]. The reaction requires external strength (high agitate or ultrasonic) to homogenize them. However, mini-emulsion polymerization usually uses C5~C6 emulsifier [65]. Fine and mini emulsion polymerization are all mixed nucleation, which can form emulsion automatically [66]. In any case, the ratio of each component and polymerization can be generated. There are also some differences in particle size.

The process of pre-emulsification in fine/mini emulsion polymerization is very critical. Usually, a stable monomer droplet is formed under the combined action of emulsifier and assistant emulsion [67], then polymerized in a submicron monomer droplet formed by high agitate or ultrasonic pre-emulsification [68]. The formation of sub-micron monomer droplets can play an electrostatic and spatial stabilizing role, thereby preventing small droplets from aggregating into large droplets. This helps reduce the content of emulsifiers to ensure the stability of the emulsion.

An emulsion of three fluoroethyl methacrylate and butyl acrylate copolymers (PF3-co-BA) was prepared by fine emulsion polymerization at Wuhan University of Technology, China [69]. The influence of reaction conditions on the nucleation size was discussed. The conversion of fluorinated monomer in polymer emulsion prepared by fine emulsion polymerization was much higher than that of conventional emulsion polymerization when the raw material ratio was the same. A kind of fluorinated acrylate copolymer (FPA) emulsion was prepared to utilize fine emulsion polymerization from the Shaanxi University of Science and Technology [70]. The latex particles in FPA emulsion are regular spheres with an average particle size of about 200 nm (Figure 2a). Chen used mini-emulsion polymerization to obtain a polymethacrylate mini-emulsion. The particle size of the emulsion was less than 100 nm. However, the large amount of emulsifier used led to its poor hydrophobicity. It can be seen that the polymer emulsion prepared by mini emulsion polymerization is usually smaller and more stable than fine emulsion polymerization. The drawback is that a large number of emulsifiers and stabilizers are needed to increase the reaction cost while the application of the emulsion is also limited.

### 3.3. Core–Shell Emulsion Polymerization

The core–shell model was first proposed in 1970 [71]. It is a multi-stage emulsion polymerization method. It takes polymerized dispersed prepolymer as the reaction core to further polymerize and finally forms a two-layer structure of core and shell. According to the way of adding monomer, it can be divided into the batch method, pre-swelling method, and semi-continuous method. Polymer emulsion with microphase separation structure [72].

Tang used a semi-continuous method to produce a core–shell fluorinated acrylate emulsion. The emulsion was based on butyl acrylate and methyl methacrylate copolymer as the core, and butyl acrylate, methyl methacrylate, and trifluoro ethyl methacrylate copolymer as the shell. The formation of the core–shell structure was confirmed by testing and analyzing the monomer conversion rate and the growth and distribution of latex particles in the emulsion polymerization. Qu et al. [73] prepared a core–shell emulsion that used silica as the core and fluorinated acrylate polymer as the shell. The effect of dodecafluoroheptyl dodecyl methacrylate and methacryloxypropyl triisoxysilane on the emulsion core–shell structure was studied. The stable core–shell emulsion can be used to prepare high-performance water-proofing materials [74]. The paint used tridecafluorooctyl methacrylate as fluorinated monomers. Two kinds of emulsions were prepared by conventional emulsion polymerization and core–shell emulsion polymerization, respectively. It was found that the hydrophobicity of the emulsions was good after coating. However, the core–shell emulsion polymerized at the same monomer concentration had better hydrophobicity. A polyurethane/fluorinated acrylate hybrid emulsion (F0) was synthesized (Figure 2b). The polymer particles in the emulsion show a regular spherical shape and have an obvious core–shell structure [75].

### 3.4. Non-Soap Emulsion Polymerization

An emulsifier is needed in conventional and fine/mini emulsion polymerization. Non-soap emulsion polymerization means that the purpose of emulsification is without an emulsifier, and uses the polarity of the reaction monomer or initiator in the system [76,77]. It is also possible to graft the polar group onto the polymer to give it a surface-active emulsion polymerization. Non-soap emulsion polymerization generally uses a reactive emulsifier, which enters the main chain of polymerization at the same time as emulsification, while ensuring the stability of the emulsion and avoiding the effect of other small molecule residual compounds on the properties of the emulsion [78]. It can also be achieved by adding cosolvent or inorganic powder. The cosolvent acts as an emulsifier and can be separated better than the conventional emulsifier. The addition of inorganic powder indicates that the particles of the emulsion are polymerized at the water/oil interface as the reaction site [79]. Non-soap emulsion polymerization is used to prepare reactive fluorinated emulsifiers and fluorinated monomers. The content of surfactant in polymer emulsion obtained by this method is very low, which can greatly enhance the surface properties of emulsion and has an unparalleled advantage in other aspects, such as optical and electrical properties.

Meng et al. [80] synthesized a dodecafluoroheptyl methacrylate fluoropolymer with fluorinated alkenes sulfonate (PSVNa) as the monomer. The results showed that 95 wt.% of the fluorinated monomer had been grafted in the main chain of the polymer, and the conversion, stability, and hydrophobicity of the emulsion were better than that polymerized by emulsifier. The pre-polymer prepared in an organic solvent can also be added to the water to disperse stable small molecule polymer emulsion, and then continue to initiate a reaction to obtain non-soap fluoropolymer emulsion. Liu et al. [81] prepared a chlorotrifluoroethylene-vinyl isobutyl ether-sodium undecanoate fluoropolymer emulsion (P(CTFE-IBVE-SUA)) by non-soap emulsion polymerization (Figure 2c). After investigating the effect of monomer ratio on polymerization, the effect of emulsified monomer SUA on emulsion and polymer performance was studied. The results showed that the polymer emulsion had strong stability and good hydrophobic ability as a coating material. The particle size distribution of the emulsion is uniform and has a core–shell structure.

It can be seen from the above methods that conventional emulsion polymerization is easy to operate and the most widely applied. Fine/mini emulsion polymerization can make the reaction easier and improve the stability of the emulsion [82,83,84,85,86]. Core–shell emulsion polymerization can show better performance under the same concentration of fluorinated monomer, and the design of the polymer molecule is easier to achieve [87,88]. Non-soap emulsion polymerization either does not use emulsifiers or uses them in a slightly superior way to emulsifier systems. The disadvantage is that fluorinated monomers with polar functional groups are usually expensive [89,90,91].

### 3.5. Structures of Fluoropolymer

In the main synthesis methods of general polymers, the synthesized polymers can be divided into homopolymers, block copolymers, and graft copolymers according to the different distributions of the monomers involved in the polymerization. They can also be divided into amorphous polymers and crystalline polymers according to the structural characteristics of the polymers [92]. Fluorinated polymers can be divided into linear, umbrella, branch, and dumbbell polymers according to their topological structure.

Different polymer structures give the prepared emulsions distinct application properties. The linear structure usually has a long fluorocarbon branched chain (Figure 3a) [93]. The main chain is the alkyl or ester group. It is used in paint as a hydrophobic anti-fouling coating material.

Compared with linear fluoropolymers, fluoropolymers with more branched chains have better performance. Figure 3b) is a fluoropolymer with an umbrella structure. In contrast experiments with linear fluorinated polymers containing the same fluorine content, it is found that the hydrophobicity of the umbrella structure emulsion is obviously improved, the contact angle between the coating interface and water is increased by 6.2°, and the water absorption of the material is also 11.1% lower than that of linear type [94].

On the basis of linear and umbrella structures, a fluoropolymer with a dendritic structure was synthesized by the Chengdu Institute of organic chemistry, Chinese Academy of Sciences [95]. The specific process of synthesis is shown in Figure 3c. The results show that the contact angle between water and dendrimer (111.3°) is higher than that of the umbrella (104.8°) and linear (98.7°) when the fluorine content is at 60%, and the fluorine content of dendrimer is 12.77% and 6.20% higher than that of linear and umbrella structure, respectively.

In addition to the above common configurations, other regular configurations of fluoropolymers can be used in specific environments and processes. In Figure 3d, Zhang et al. [96] prepared dumbbell-shaped fluoro-graft polymers (FMCDs) with polyglycidyl ether-PEG (PBG) as the reaction core, and then polymerized with fluorinated monomers after activation. The polyether structure in the main chain made it highly water-soluble. The fluoromethyl group at the end of the chain showed extremely strong hydrophobicity, with good performance in the emulsification and demulsification process. The design and development of such structures can provide new research ideas for the research and application of fluoropolymers.

The above-fluorinated polymers are symmetrical and regular in structure, all of which show the related properties of the crystal. In addition, there are amorphous fluorinated polymers. This can be seen from the fluorinated polymers with the amorphous structure shown in Figure 3e [97]. The fluorinated functional groups on the surface are more interlaced with each other than the previous fluorinated polymers. Due to the special structure of raw materials, the research cost is high, the reaction controllability is poor, and the related application examples are also limited. It could certainly be applied in the fields of super bispecific materials and multiphase separation [98].

Combining the above examples of fluoropolymer with different topological structures, the fluoropolymer emulsion is used as the interface material. Its hydrophobicity/oleophobicity performance mainly comes from the fluorocarbon chain of the fluoropolymer. The closer the interface arrangement is, the longer the fluorine chain is, and the higher the fluorine content of the surface is, the better the hydrophobicity/oleophobicity performance is.

## 4. Wettability Application

### 4.1. Interaction with Nanomaterials

As the frontier science of materials research, nanomaterials can significantly improve the performance of materials with their unique properties of nano size. Fluoropolymer materials can be combined with nanomaterials to prepare nano fluoropolymer materials, so as to enhance or improve the properties of materials and make the interface achieve the excellent performance of hydrophobicity or super-amphiphilicity [99]. The nano-grade fluoropolymer emulsion can be directly synthesized through polymerization, and can also be prepared by grafting modification with nanomaterials. At present, the preparation method is more economical and the fluorinated material is used to modify the surface of nano silica. The nanomaterials obtained by this method are stable in performance, uniform in particle size, easy to operate and prepare, and extensive in raw materials.

Figure 4a briefly describes a method of preparing fluorinated nanomaterials from fluorinated polymer and nano-SiO_2_ [100]. This method takes nano-SiO_2_ prepared by tetraethoxysilane (TEOS) as the core and modifies fluorinated ethoxy silane to obtain a fluorinated nanomaterial. It has super liquid rejection performance. The contact angle with various liquids is more than 140° after being prepared into a film (water: 167.5°, diesel: 140.4°, soybean oil: 146.6°, xylene: 140.5°, diiodomethane: 158.6°, decalin: 142.5°). Jiang et al. [101] synthesized a fluorinated block copolymer and attached it to the surface of nano-SiO_2_ (Figure 4b). The obtained fluorinated polymer nanoparticles have a raspberry structure and also show good amphiphilic properties in terms of properties. The contact angle between the prepared interface and water and oil exceeds 150 °C. Figure 4c is the scanning electron microscope results of nanomaterials with different fluorine content prepared by changing the concentration of fluorinated monomers and nanoparticles [102]. The results show that the larger the fluorine content of the system is, the smaller the particle size of the prepared nanoparticles is, and the better the performance is. The contact angle with water and n-dodecane is about 180° and 109°, respectively.

In addition to the nano-SiO_2_ system, other nanomaterials such as graphene, nano-cellulose, nano-coal ash, and some inorganic nano-oxides can be used to interact with fluoropolymers. Adam mixed fluoropolymer emulsion and nano-ZnO particles to prepare a composite nano spray material. In Figure 4d, the change of the contact angle of the nanomaterial prepared by the mixture of acetone and water in the mixing agent was studied [103]. The results show that the prepared nanomaterials have good hydrophobicity at different cosolvent ratios, and the contact angle with water (oil) droplets can reach 160° at a certain cosolvent ratio.

### 4.2. Application on EOR

With the change in national energy strategy, especially in the progress of enhancing oil recovery (EOR), the exploration and exploitation of oil and gas play an important strategic role. The ability of fluoropolymer emulsion to improve and transform the wettability of the interface can be applied to the transformation of reservoir wettability, the development of condensate gas reservoirs, and the injection and reflow process.

Feng et al. [104] obtained a fluorinated two-affinity block polymer emulsion using polyethylene glycol monomethyl ether (MPEG) and dodecafluoroheptyl methacrylate as the main raw materials. Figure 5a shows the wettability of the core surface on different droplets before and after treatment with fluoropolymer emulsion. By measuring the contact angle, the wettability of the core surface is changed from parent to gas wetting, indicating that the polymer emulsion can be used as a wetting reversal agent to change the wettability of the reservoir.

Similarly, a fluoropolymer emulsion prepared by Jin et al. [105] was used as a wetting reversal agent to treat the core. At the concentration of 0.1 wt.%, the contact angle between the core with water and n-hexadecane increased from 23° and 0° before treatment to 61° and 70° after treatment. When the concentration was at 0.3 wt.%, the contact angles increased to 152° and 127°, respectively. It can be seen from Figure 5b that the fluorinated polymer modified by nano-SiO_2_ on the core surface has a stronger wettability reversal ability after adhesion. It can be used to improve the wettability of the core. The wettability of the core changes from liquid wetting to strong gas wetting.

Our team has also researched fluoropolymer emulsion as a wetting reversal agent. Figure 5c is a polymer microemulsion wetting reversal agent (PFG4) [106]. The emulsion has strong stability, dispersibility, and very low surface free energy. After injection, it can effectively change the wettability of the formation and increase the permeability of the formation to achieve the effect of enhancing oil recovery and flow back rate and reducing the resistance of oil and gas production. In summary, the application of fluoropolymer emulsion is mainly used in the presence of a solid–liquid interface. As a coating material or nano adsorbent material, it can act on the solid surface and change the wettability of the solid surface by reducing the interfacial tension and achieving the effect of hydrophobicity or super-amphiphobicity.

## 5. Conclusions and Outlook

With the continuous in-depth study of fluoropolymer emulsion, not only have new methods of preparing such functional materials been put forward, but also the practical application and mechanism of action in various fields have been further explored and expanded. At present, the research focus of fluoropolymer emulsion is to enhance the performance of fluoropolymer and reduce the cost of synthesis and harm to the environment. Therefore, the research of fluoropolymer emulsion has a good application prospect.

Fluorinated acrylate polymers, as the main research objects of fluoropolymer emulsion, have shown excellent interfacial properties for a long time. The conditions of synthesis and source of raw materials have been relatively stable and mature. However, a large number of non-emulsifiers exist widely in these emulsions. Non-soap emulsion polymerization can avoid adverse effects caused by emulsifiers. The monomer synthesis required for the reaction is difficult, expensive, and difficult to preserve. Therefore, improving the synthesis methods of such monomers to reduce the cost of preparation, improving the emulsifying system, and introducing a system are other directions of fluoropolymer emulsion.

In application, fluoropolymer emulsion is mainly used as coating material, synthesis of nanoparticles, or modification with nanomaterials to improve the hydrophobicity ability of solid surfaces. It can also be used as a polymer surfactant with appropriate concentration. It may also be extended to take into account experiments and computational models showing that the interfacial properties of polymers can be adjusted by varying the composition of polymer mixtures with two different polymer topologies [107,108,109,110]. However, in the application process, the circulation of fluorine and its harm to the environment is only slightly mentioned. There is no detailed quantitative analysis and research, which can be used as a new research direction.

## Figures and Tables

**Figure 1 molecules-28-00905-f001:**
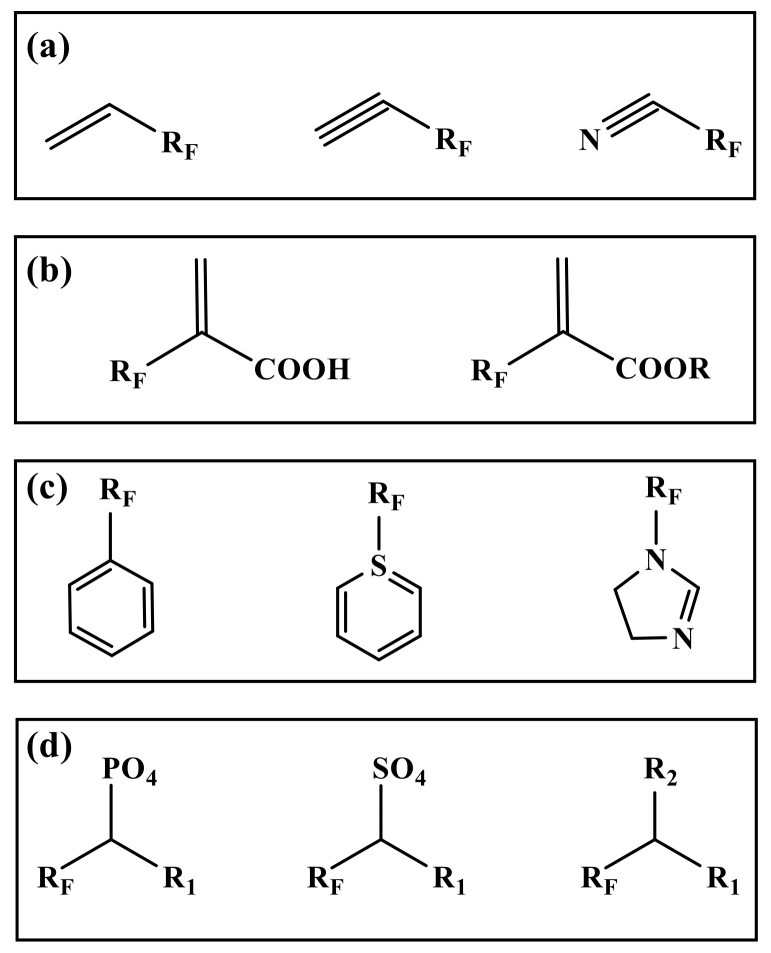
Structural diagram of several types fluorinated monomers. (**a**): Alkene (Alkyne) fluorinated monomers, (**b**): Acrylic acid (ester) fluorinated monomers, (**c**): Fluorinated aromatic (heterocyclic) monomers, (**d**): Others. (R for alkyl group. R_F_ for fluoroalkyl. R_1_, R_2_ for halogen atom, hydroxyl group, amino group, etc.).

**Figure 2 molecules-28-00905-f002:**
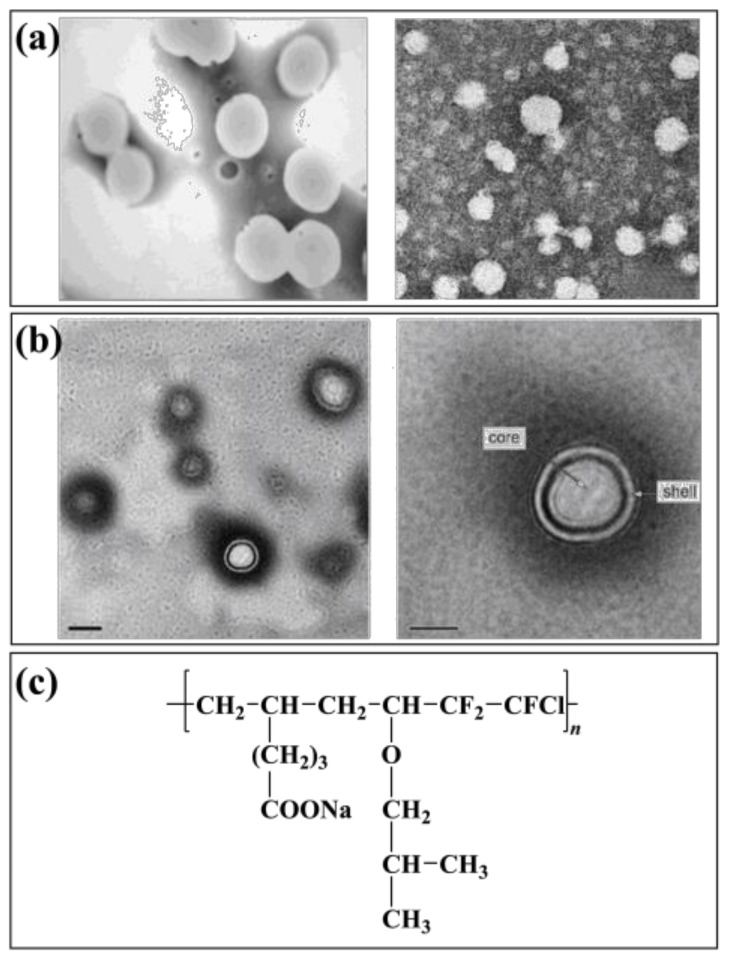
Fluoropolymer and emulsion polymerization (**a**): TEM of FPA emulsion, (**b**): TEM of F0 emulsion, (**c**): Structure diagram of P(CTFE-IBVE-SUA).

**Figure 3 molecules-28-00905-f003:**
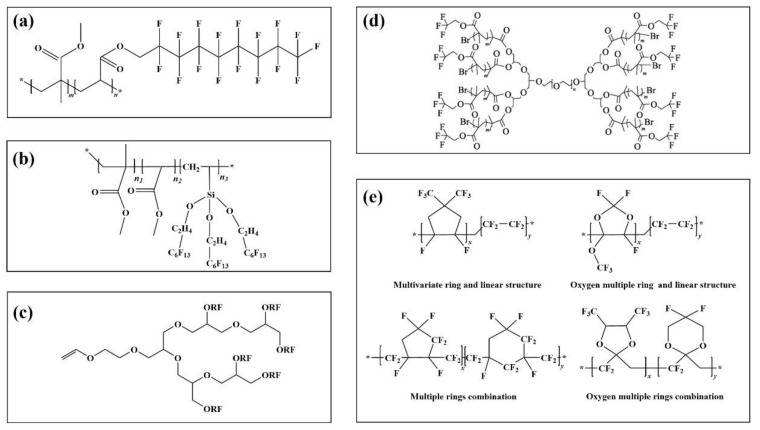
Fluoropolymers with different configurations (**a**): a linear fluoropolymer, (**b**): a dendrimer fluoropolymer, (**c**): an umbrella fluoropolymer, (**d**): a dendrimer fluoropolymer, (**e**): some amorphous fluoropolymers.

**Figure 4 molecules-28-00905-f004:**
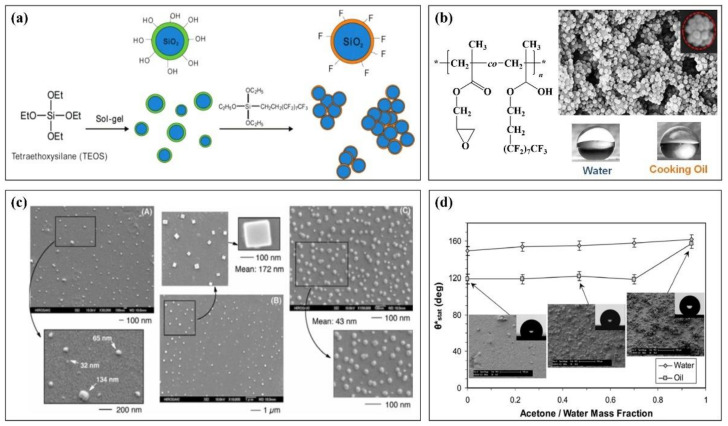
Application of fluoropolymer in coating nanomaterials (**a**): Preparation method of nano-silica particles of a fluoropolymer, (**b**): Molecular structure of a fluoropolymer and the SEM of its raspberry nanomaterials, (**c**): SEM of the surface of the nanomaterial with different fluorine content, (**d**): Apparent contact angle of oil and water droplets as a function of acetone cosolvent concentration.

**Figure 5 molecules-28-00905-f005:**
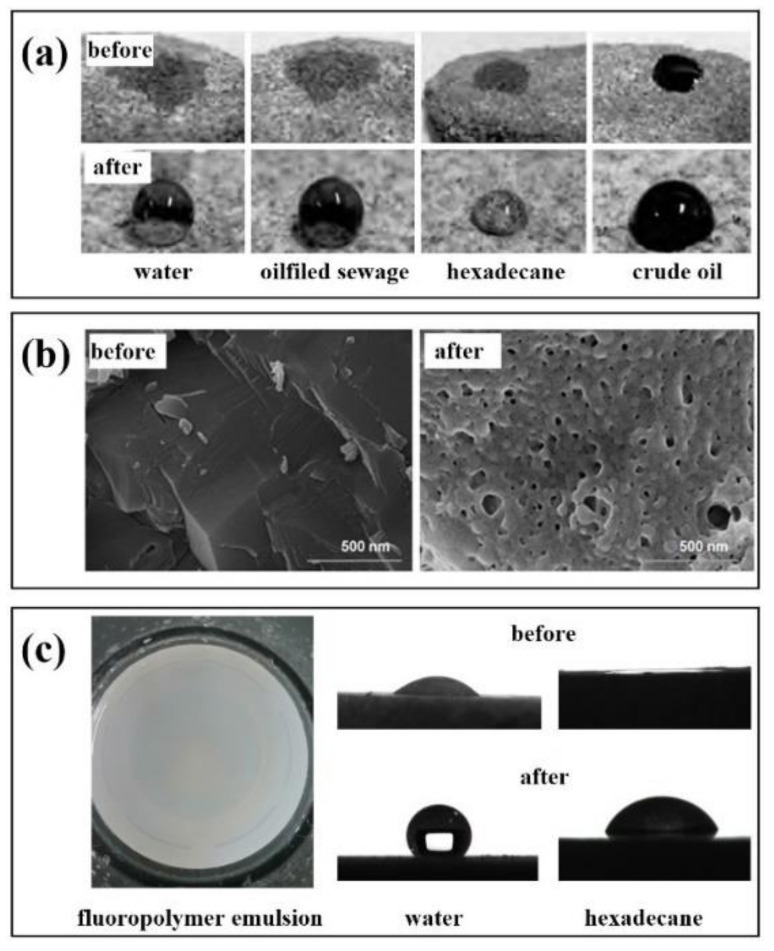
Application of fluoropolymer in reservoir wettability reversion (**a**): Contact angle of different droplets with the cores, (**b**): SEM of the cores with gas-wetted nano-SiO_2_, (**c**): Polarizing micrographs and apparent morphology of PFG4 emulsion.

## Data Availability

Not applicable.

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
