# Peer review of "Fluoropolymer: A Review on Its Emulsion Preparation and Wettability to Solid-Liquid Interface"

_molecules, 2023, doi:10.3390/molecules28020905_

Round 1

Reviewer 1 Report

Lei Liang, et al. reported a review on "Fluoropolymer: A Review on Its Emulsion Preparation and  Wettability to Solid-Liquid Interface", the subject of the paper is worth publishing in Molecules, but many grammatical and syntax errors make reading difficult, and the paper should be considered for REVIEW ONLY AFTER a complete revision of the English language. The paper should be rejected and reconsidered after the language revision. Besides, bibliographic citations should be added to the figures.

Author Response

Dear Editors and Reviewers:

Thank you for your letter and for the reviewers’ comments concerning our manuscript entitled “Fluoropolymer: A Review on Its Emulsion Preparation and Wettability to Solid-Liquid Interface” (Manuscript ID: molecules-2042152). Those comments are all valuable and very helpful for revising and improving our paper, as well as the important guiding significance to our research. We have studied the comments carefully and have made corrections which we hope meet with approval. Revised portions were marked in “Track Changes” in the manuscript.

The main corrections in the paper and the responses to the reviewer’s comments are as flowing:

Responds to the reviewer’s comments:

Reviewer 1:

Concern # 1: The paper should be considered for review only after a complete revision of the English language.
Response: We have polished the manuscript comprehensively. In especially, the grammar and spelling were carefully corrected. We also have used an editing service to increase the readability of the paper. The “Language Revision Certificate” was provided as an attached file to the Editorial Office, and we sincerely hope that the reviewer will reconsider accepting our manuscript.

Concern # 2: Besides, bibliographic citations should be added to the figures.
Response: Bibliographic citations were added after the title of all cited figures.

Special thanks to you for your good comments.

We tried our best to improve the paper and made some changes to the manuscript. These changes will not influence the content and framework of the paper. We appreciate for Editors/Reviewers’ warm work earnestly and hope that the correction will meet with approval.

Once again, thank you very much for your comments and suggestions.

Reviewer 2 Report

The authors reviewed the preparation and application progress in fluoropolymer emulsion and put forward some thoughts. The work is interesting and might be helpful for further development of fluoropolymer emulsion. It may be publishable after the appropriate revision.

Comments:

1.      The abstrate should be more concise and readable.

2.      In the abstract, it was announced that the C-F bond has the lowest surface free energy in all chemical bonds, however, in Line 31-32 it was written that the C-F bond has a very low free energy. Please check it and cite relevent important papers to support the statement.

3.      Please improve the English of the manuscript and make it more logically. For example, it seemed not easy for the reader to catch the main idea of Paragraph 1 in the Introduction. Another example is: Line 45-46: According to the dispersion medium, it can be divived into water-based emulsion and oil-based emulsion. What is “it” in the sentence?

4.      Line 48-49: It seems better to complete all lists rather than “other fields” for  which the continuations will not be obvious to the general readership.

5.      Line 95-96: It is better to give the name of the functional groups.

6.      Line 103: Please list the name of several types.

7.      Could the authors give more introduction of these monomers?

8.      Line 172-184: It seems better to introduce the scholars’ work in a more logic way.

9.      Line 363-373: Attention should be paid to the precision of contact angles. In my opinion, it is better to use 159 degrees with a standard error rather than 158.6 degrees. Is the surface smoothly flat or rough? If it is a rough surface, what is the effect of surface roughness on wettability of the surfaces made of fluoropolymer? Moreover, what is the contact angle hysteresis?

Author Response

Reviewer 2:

Concern # 1: The abstract should be more concise and readable.
Response: We have carefully revised the abstract section.

Concern # 2: In the abstract, it was announced that the C-F bond has the lowest surface free energy in all chemical bonds, however, in Line 31-32 it was written that the C-F bond has a very low free energy. Please check it and cite relevent important papers to support the statement.
Response: We apologize for the inaccuracy statement and changed it as follows: “the C-F bond has a very low surface free energy”.

Concern # 3: Please improve the English of the manuscript and make it more logically. For example, it seemed not easy for the reader to catch the main idea of Paragraph 1 in the Introduction. Another example is: Line 45-46: According to the dispersion medium, it can be divived into water-based emulsion and oil-based emulsion. What is “it” in the sentence?
Response: We have polished the manuscript comprehensively. In especially, the grammar and spelling were carefully corrected. We also have used an editing service to increase the readability of the paper. In the sentence, “it” refers to “fluoropolymer emulsion”.

Concern # 4: Line 48-49: It seems better to complete all lists rather than “other fields” for which the continuations will not be obvious to the general readership.
Response: We can’t list all the application fields, so the “other fields” is an extension of the list.

Concern # 5: Line 95-96: It is better to give the name of the functional groups.
Response: We have added examples of reactive functional groups at corresponding positions.

Concern # 6: Line 103: Please list the name of several types.
Response: The specific types were explained in the paper.

Concern # 7: Could the authors give more introduction of these monomers?
Response: The monomer introduction is only a small part of the paper, so there is no further in-depth discussion.

Concern # 8: Line 172-184: It seems better to introduce the scholars’ work in a more logic way.
Response: We enumerate and analyze the research results prepared by the method of “Conventional emulsion polymerization”.

Concern # 9: Line 363-373: Attention should be paid to the precision of contact angles. In my opinion, it is better to use 159 degrees with a standard error rather than 158.6 degrees. Is the surface smoothly flat or rough? If it is a rough surface, what is the effect of surface roughness on wettability of the surfaces made of fluoropolymer? Moreover, what is the contact angle hysteresis?
Response: The contact angle values here were extracted from the original data of the cited articles. The wettability of the interface is determined by its free energy of the interface. The free energy of an interface can also be reformed by changing the roughness of the surface. If a surface is chemically hydrophobic, it will become more hydrophobic when the surface roughness is increased, and vice versa.

Special thanks to you for your good comments.

We tried our best to improve the paper and made some changes to the manuscript. These changes will not influence the content and framework of the paper. We appreciate for Editors/Reviewers’ warm work earnestly and hope that the correction will meet with approval.

Once again, thank you very much for your comments and suggestions.

Reviewer 3 Report

This review focuses on fluoropolymer emulsions, which are materials with interesting coating properties, also used as building and medical materials and in conjunction with nanoparticles to enhance hydrophobicity and then wettability at solid surfaces. I found the review useful and informative, written with an easy and pleasant narrative. My very minor comment is related to the use of fluorinated acrylate copolymers, based on latex particles or dodecafluoroheptyl methacrylate, to obtain polymethacrylate mini emulsions. The literature may also be extended to take into account experiments and computational models showing that interfacial properties of polysterene polymers can be adjusted by varying the composition of polymer mixtures with two different polymer topologies (Phys. Rev. Lett. 111, 068303 (2013); Phys. Rev. E, 93, 050501 (2016); Phys. Rev. Lett. 118, 167801 (2017)).

Author Response

Dear Editors and Reviewers:

Thank you for your letter and for the reviewers’ comments concerning our manuscript entitled “Fluoropolymer: A Review on Its Emulsion Preparation and Wettability to Solid-Liquid Interface” (Manuscript ID: molecules-2042152). Those comments are all valuable and very helpful for revising and improving our paper, as well as the important guiding significance to our research. We have studied the comments carefully and have made corrections which we hope meet with approval. Revised portions were marked in “Track Changes” in the manuscript.

The main corrections in the paper and the responses to the reviewer’s comments are as flowing:

Responds to the reviewer’s comments:

Reviewer 3:

Concern # 1: The literature may also be extended to take into account experiments and computational models showing that interfacial properties of polysterene polymers can be adjusted by varying the composition of polymer mixtures with two different polymer topologies (Phys. Rev. Lett. 111, 068303 (2013); Phys. Rev. E, 93, 050501 (2016); Phys. Rev. Lett. 118, 167801 (2017)).
Response: Taking into account the comments of the reviewer, the citations were added and the references in the paper were also updated.

Special thanks to you for your good comments.

We tried our best to improve the paper and made some changes to the manuscript. These changes will not influence the content and framework of the paper. We appreciate for Editors/Reviewers’ warm work earnestly and hope that the correction will meet with approval.

Once again, thank you very much for your comments and suggestions.

Round 2

Reviewer 1 Report

The manuscript is now worth the publication in molecules.

Reviewer 2 Report

The authors have satisfactorily answered my comments and I recommend the manuscript for publication in Molecules.